# Coronavirus Disease 2019-Associated Thrombotic Microangiopathy: Literature Review

**DOI:** 10.3390/ijms231911307

**Published:** 2022-09-25

**Authors:** Marija Malgaj Vrečko, Andreja Aleš Rigler, Željka Večerić-Haler

**Affiliations:** 1Department of Nephrology, University Medical Center Ljubljana, 1000 Ljubljana, Slovenia; 2Faculty of Medicine, University of Ljubljana, 1000 Ljubljana, Slovenia

**Keywords:** thrombotic microangiopathy, thrombotic thrombocytopenic purpura, atypical hemolytic uremic syndrome, COVID-19, acute kidney injury

## Abstract

Coronavirus disease 2019 (COVID-19) can lead to clinically significant multisystem disorders that also affect the kidney. According to recent data, renal injury in the form of thrombotic microangiopathy (TMA) in native kidneys ranks third in frequency. Our review of global literature revealed 46 cases of TMA in association with COVID-19. Among identified cases, 18 patients presented as thrombotic thrombocytopenic purpura (TTP) and 28 cases presented as atypical hemolytic uremic syndrome (aHUS). Altogether, seven patients with aHUS had previously proven pathogenic or likely pathogenic genetic complement abnormalities. TMA occurred at the time of viremia or even after viral clearance. Infection with COVID-19 resulted in almost no or only mild respiratory symptoms in the majority of patients, while digestive symptoms occurred in almost one-third of patients. Regarding the clinical presentation of COVID-19-associated TMA, the cases showed no major deviations from the known presentation. Patients with TTP were treated with plasma exchange (88.9%) or fresh frozen plasma (11.1%), corticosteroids (88.9%), rituximab (38.9%), and caplacizumab (11.1%). Furthermore, 53.6% of patients with aHUS underwent plasma exchange with or without steroid as initial therapy, and 57.1% of patients received a C5 complement inhibitor. Mortality in the studied cohort was 16.7% for patients with TTP and 10.7% for patients with aHUS. The exact role of COVID-19 in the setting of COVID-19-associated TMA remains unclear. COVID-19 likely represents a second hit of aHUS or TTP that manifests in genetically predisposed individuals. Early identification of the TMA subtype and appropriate prompt and specific treatment could lead to good outcomes comparable to survival and recovery statistics for TMA of all causes.

## 1. Introduction

Since December 2019, when the first patients were diagnosed with severe acute respiratory syndrome coronavirus 2 (SARS-CoV-2) infection, coronavirus disease 2019 (COVID-19) has spread tremendously. We now know that, in addition to pulmonary involvement, infected individuals may present with a wide range of extrapulmonary manifestations, including acute kidney injury (AKI), thrombotic complications, myocardial insufficiency and arrhythmias, acute coronary syndromes, gastrointestinal symptoms, hepatocellular injury, hyperglycemia and ketosis, neurologic disease, ocular symptoms, and dermatologic complications [1]. AKI is common in patients with COVID-19 [2], but its exact mechanism remains unclear. While acute tubular injury appears to be the most common histopathologic finding on renal biopsy [3,4], it has been suggested that thrombotic microangiopathy (TMA) can also occur [5].

### Basic Concepts of Definition, Classification, Diagnosis, and Treatment of TMA

TMA is a group of extraordinarily diverse syndromes that can be inherited or acquired and can occur suddenly or gradually in children or adults [6]. Despite their diversity, the TMA syndromes share certain pathological and clinical features. The most important pathological feature is widespread thrombosis in capillaries and arterioles, clinically manifested by microangiopathic hemolytic anemia (MAHA), thrombocytopenia, and organ damage (AKI, neurological abnormalities) [6]. Occasionally, TMA may manifest with accelerated arterial hypertension and impaired renal function with only mild, barely perceptible MAHA. In these cases, the disease can be detected by renal biopsy [6,7].

Classification of TMA is difficult and evolving, with recent consideration of the impact of genetic background and etiologic triggers [8,9,10,11]. For the purposes of this review article, we use the classification proposed by Goodship et al. [11], which divides TMA into the following forms: thrombotic thrombocytopenic purpura (TTP) resulting from an inherited or acquired deficiency of ADAMTS13; typical hemolitic uremic syndrome caused by Shiga toxin-producing *Escherichia coli* (STEC-HUS); all other forms grouped under the common term atypical HUS (aHUS). aHUS can be either primary/hereditary with uncontrolled activation of the alternative complement pathway (due to a pathogenic complement gene mutation) or secondary/acquired. In fact, all conditions previously categorized as secondary TMA associated with a spectrum of diseases (TMA associated with pregnancy, transplantation, drugs, bone marrow transplantation, autoimmune diseases, malignancies, glomerular diseases such as ANCA-associated vasculitis, IgA nephropathy, HELLP, and TMA with severe hypertension) are now grouped under the common denominator of aHUS. The pathogenic mechanisms in these secondary forms of aHUS are multifactorial, and complement system abnormalities representing a “first hit” are thought to be unidentified in many of these patients. The classification into primary/hereditary and secondary/acquired TMA is not absolute, as hereditary TMA requires a trigger factor, and acquired TMA may also have a genetic background [12,13]. COVID-19-associated TMA represents a new, distinct entity that may present as TTP or aHUS and probably deserves a special, additional place in the classification.

The diagnosis of TMA is made on the basis of clinical features (MAHA, thrombocytopenia, and organ damage) and additional diagnostic tests to further specify the nature of TMA. An ADAMTS13 activity level of less than 10% supports a diagnosis of TTP. However, diagnosis of TTP is difficult because ADAMTS13 testing is limited, especially in the COVID-19-associated pandemic that is putting pressure on hospitals and laboratories. Clinical prediction scores (e.g., PLASMIC score, French score, and Bentley score) have been developed to estimate the likelihood of severe ADAMTS13 deficiency with high sensitivity and specificity and, together with clinical assessment, can guide initial treatment decisions [14]. In the heterogeneous group of aHUS, the underlying cause should be determined, such as the presence of malignancy, pregnancy, autoimmune disease, etc. In primary aHUS, genetic abnormalities such as inactivating mutations in genes encoding complement-regulating proteins or gain-of-function mutations in genes encoding complement-activating proteins are found in about half of the patients [15]. Unfortunately, there is no diagnostic test that unequivocally confirms primary aHUS, and the diagnosis is considered a diagnosis of exclusion.

Treatment of TMA generally relies on four modalities: therapeutic plasma exchange (TPE), immunosuppression, monoclonal antibodies, and treatment of the underlying cause. Historically, TPE in combination with glucocorticoids was considered the only option for the treatment of TMA. Although we still cannot fully explain the pathophysiologic effect, TPE has been shown to be beneficial also in the treatment of other types of TMA (not just TTP) [16]. Part of its effect is certainly to replace the defective or deficient protein (such as ADAMTS13, complement, etc.) with a functional one [17]. Glucocorticoids have a versatile mechanism of action: inhibition of activation of the alternative complement pathway [18], inhibition of acquired ADAMTS13 inhibitors (in acquired TTP) and anti-factor H antibodies (in acquired aHUS) [19], and suppression of endothelial inflammation [20]. Other immunosuppressants have also been used to treat TMA, including the B-cell-depleting monoclonal antibody rituximab, which has been shown to reduce clinical relapse in acquired TTP [21]. Specific diagnosis of TMA type has become more important since the discovery of terminal complement inhibitors such as eculizumab or ravulizumab. These monoclonal antibodies bind to complement protein C5 and allow targeted treatment of complement-mediated forms of TMA [22]. Recently, a targeted TTP treatment was approved, i.e., caplacizumab. This is a humanized bivalent anti-vWF immunoglobulin fragment that inhibits the interaction between vWF multimers and platelets [23].

## 2. COVID-19-Associated TMA

It has become clear that TMA can emerge in association with SARS-CoV-2 infection [5,24]; however, the exact role of COVID-19 in the setting of these TMAs remains unclear.

To gain broader insight into the global knowledge of COVID-19-associated TMA, we reviewed the global literature reported from the first mention of the disease in 2019 to February 2022, including individual case reports and case series of adult patients with emphasis on acute kidney injury in the setting of these TMAs. Electronic searches were conducted in the PubMed database using the following keywords and their combinations: thrombotic microangiopathy, COVID-19, thrombotic thrombocytopenic purpura, and atypical hemolytic uremic syndrome.

The search revealed 46 cases of TMA in association to COVID-19 in adult patients, which are collected in Table 1 and Table 2. Certain cases [25,26,27,28] were excluded due to missing data or to the fact that the presented case also received a vaccine against COVID-19 prior to the occurrence of TMA.

### 2.1. Types of TMA Associated with COVID-19 and Clinical Correlates

Using the described search criteria, we found 46 published cases of TMA associated with COVID-19 from the time COVID-19 was first mentioned until February 2022. On the basis of the Goodship et al. classification used for this analysis, 18 cases were presented as TTP, and a total of 28 cases were presented as HUS. Of the patients with COVID-19 associated HUS, none were associated with Shiga toxin, whereas there were seven patients with primary aHUS, and another 21 patients in this group classified as secondary aHUS (associated with various causes, see Table 2).

#### 2.1.1. Patients Presenting as TTP

In patients presenting as TTP, antibodies to ADAMTS13 were detected in all but three cases (in which ADAMTS13 antibody evaluation was not available), and ADAMTS13 activity was reported as <10% in all reported cases. In three cases [35,36,41] in which TTP was reported as recurrent, no further investigations or genetic testing were reported. Patients with the clinical picture of COVID-19-associated TTP had a median platelet count of 14 (range 5–100), and nine patients had fully recovered renal function, or it had been normal since the onset of the disease. Renal biopsy was not performed in any of the cases.

Of the 18 TTP patients, seven had no known concomitant diseases, and one woman in this group was pregnant. Reported comorbidities included autoimmune diseases, i.e., Crohn’s disease and SLE with antiphospholipid syndrome (APLS) (*n* = 2), history of breast cancer (*n* = 2), hypertension (*n* = 3), type 2 diabetes mellitus (*n* = 1), and peripheral arterial disease (*n* = 1), and two patients were overweight.

The clinical presentation of COVID-19-associated disease in patients with TTP was described as mild in 16 cases (no need for intensive care, no need for ventilatory support). However, in addition to mostly mild or moderate respiratory symptoms and typical dysgeusia and anosmia, symptoms and signs related to the gastrointestinal tract were common (*n* = 5), including nausea, vomiting, diarrhea, and abdominal pain.

The majority of patients (*n* = 17) had a positive polymerase chain reaction (PCR)-based smear for COVID-19 at the time of TTP presentation, and one patient had typical respiratory symptoms with a negative PCR smear but positive serology at the time of TTP presentation. One patient with a negative PCR smear but positive serology had developed TTP 1 month after respiratory symptoms.

Patients with TTP were treated with TPE/FFP alone (*n* = 2) or in combination with steroid (*n* = 6). Altogether, seven patients received rituximab, and five patients were also treated with caplacizumab.

Of 18 patients who developed TTP, 15 recovered, including complete recovery of renal function in seven of those reported. In another three patients, renal function was intact from the onset of clinical presentation, whereas renal function data were not available in the remaining patients. Patient outcomes in relation to therapy received are shown in Table 3.

Three deaths were reported in patients with TTP: A 56 year old woman with a history of breast cancer treated with TPE and rituximab [30], a 69 year old man with a history of multiple TTP relapses treated with TPE and steroids [35], and a 21 year old man with no known comorbidities treated with fresh frozen plasma [44].

#### 2.1.2. Patients Presenting as aHUS

According to the classification criteria proposed by Goodship et al. [11], we could classify another 28 reported cases as aHUS. According to this classification, three of the reported cases had relapsing disease (presumed primary aHUS) with previously diagnosed pathogenic genetic complement abnormalities: a heterozygous pathogenic variant in the membrane cofactor protein-encoding gene in one case [45], a C3 gene mutation [47], and a heterozygous variant of unknown significance in CFI with risk polymorphisms in CFH. [50] In three patients with no previously known aHUS, there were likely pathogenic complement gene variants associated to an altered regulation of the complement alternative pathway detected, and one patient had the risk haplotype (H3) in the CFH gene, which predisposes to aHUS [46].

An additional seven patients with aHUS [48,49,51,52,53,61] showed laboratory evidence of functional dysregulation of the alternative complement pathway, although no genetic analysis of the complement system was performed in all but one [61]. No complement abnormalities were detected in the other 14 reported aHUS patients.

It is well known that many patients with a clinical presentation of aHUS (regardless of a recognized underlying complement risk factor) usually require a trigger for aHUS to manifest. Among the reported conditions that could act as a triggering factor for the onset of aHUS, there are only five patients in whom no preliminary event or evident coexisting pathologies were reported, and COVID-19 appeared to be the only recognized trigger for aHUS [48,49,52,61,64], whereas, in all other cases, various other triggering factors were observed in addition to COVID-19 infection. Eight of the reported patients were solid organ transplant recipients (heart or kidney) treated with immunosuppressants known to be associated with TMA (e.g., calcineurin inhibitors (CNI) and sirolimus), one patient was treated with gemcitabine for disseminated cancer, and the other recognized risk factors included concurrent treatment with hydroxychloroquine (HCQ), pregnancy, chronic infections such as HCV, and several others.

All but seven patients with aHUS had known comorbidities. Hypertension was present in seven patients, and seven patients were solid organ recipients with end-stage renal disease (ESRD) due to Liddle syndrome, primary focal segmental glomerulosclerosis (FSGS), nephroangiosclerosis, and IgA nephropathy, while the sole heart transplant recipient also had childhood leukemia. One patient suffered from asthma, and another suffered from liver cirrhosis related to HCV infection. One patient was classified as obese.

The clinical presentation of COVID-19 associated respiratory disease was reported as severe-with need for intensive care or ventilatory support (*n* = 2), mild to moderate (*n* = 15), or without respiratory symptoms (*n* = 8). A total of eight patients also experienced symptoms or clinical signs related to the gastrointestinal tract, including diarrhea, abdominal pain, dysphagia, vomiting, and acute pancreatitis.

Only one patient with underlying abnormalities of the alternative complement pathway [61] experienced aHUS after clearance of the virus (negative PCR smear but positive serology for COVID-19). In all other patients, the PCR smear was positive at the time of aHUS presentation.

With the exception of one patient in whom data were missing, all other patients with aHUS experienced worsening renal function. A total of 18 patients with aHUS recovered renal function after treatment (partial recovery, *n* = 6; complete recovery, *n* = 9; recovery of unknown extent, *n* = 3), whereas seven patients experienced end-stage renal failure. Patients with the clinical picture of COVID-19-associated aHUS had a median platelet count of 36 (range 6–157). Renal biopsy was performed in 14 patients, all of whom had pathohistological signs of TMA.

Three deaths were reported in patients with aHUS: A 69 year old woman with asthma and mild respiratory symptoms associated with COVID-19 treated with eculizumab [53], a 66 year old woman with severe respiratory disease including ARDS treated with fresh frozen plasma, tocilizumab, and eculizumab [49], and an 81 year old woman who died before treatment [62]. Stillbirth also occurred in a pregnant 21 year old woman (who was also amphetamine-dependent), but she recovered fully after labor and TPE [60].

Patients with aHUS were treated with TPE or FFP alone (*n* = 5) or in combination with steroid (*n* = 1). Steroid alone was used in one patient, whereas 16 patients received a C5 complement inhibitor. Patient outcomes in relation to therapy received are shown in Table 3.

## 3. Discussion

COVID-19 can lead to clinically significant multisystem disorders that also affect the kidney. To date, evidence for COVID-19-associated kidney disease has come from in vivo or postmortem studies of native or transplanted kidney biopsies from patients who had COVID-19. According to data from one of the most recent meta-analyses performed, renal injury in the form of TMA in native kidneys ranked third in frequency, followed by collapsing FSGS and acute tubular injury [66]. To provide a comprehensive review of the clinical and histopathological features described in COVID-19-associated TMA, we performed an analysis of previously reported cases and case series of patients affected by COVID-19. The review identified a total of 46 cases of COVID-19-associated TMA during the period from the onset of the COVID-19 pandemic in 2019 to February 2022.

The cases presented here show typical features suggestive of systemic TMA (including MAHA with platelet consumption and organ dysfunction) that occurred at the time of COVID-19 viremia but also relatively later, up to 1 month after viral clearance. Interestingly, infection with COVID-19 resulted in almost no or only mild respiratory symptoms in majority of both TTP and aHUS patients. Noteworthily, digestive symptoms occurred in almost one-third of patients. Although COVID-19 is primarily a respiratory disease, there are now increasing reports of subgroups of COVID-19 patients in whom gastrointestinal symptoms predominate, particularly diarrhea, anorexia, vomiting, and nausea, or who have only gastrointestinal clinical signs in the absence of respiratory symptoms [67]. The possible relationship between the clinical presentation of COVID-19 infection and the occurrence of TMA is still unclear. However, it was found that patients with digestive symptoms presented later for treatment than patients with respiratory symptoms, had a longer time interval between the onset of symptoms and viral clearance, were more likely to test positive for fecal virus only [68], and had a longer clotting time and higher liver enzyme levels [69]. Whether this is in any way related to or poses risk for TMA in patients with COVID-19 remains unclear.

On the basis of differentiated clinical and laboratory parameters (renal function tests, coagulation tests, ADAMTS13 levels, PLASMIC score, and detected complement abnormalities), COVID-19-associated aHUS is somewhat more common in the literature published to date. This is surprising, as the relative frequency of TTP and aHUS in patients with TMA generally shows a much higher proportion of aHUS (i.e., up to six times more aHUS is reported compared with TTP) [70]. Possible reasons for this are that many cases of COVID-19-associated aHUS are likely to have been overlooked, as it may have occurred in some patients with a significant time delay after COVID-19 infection and is also associated with a less “extreme” clinical presentation in aHUS variants (e.g., fewer or no neurologic abnormalities), which may then be underrepresented.

Regarding the clinical presentation of COVID-19-associated TMA, the cases presented here show no major deviations from the known presentation, in which patients with TTP usually present MAHA with severe thrombocytopenia, frequent neurological symptoms, and multiple organ involvement, with renal failure being rare. In contrast, thrombocytopenia in patients with aHUS is usually moderate, the disease is often confined to the kidneys, and renal failure is consequently more common [71].

Patients with the clinical picture of COVID-19-associated TTP were characterized by lower platelet counts (median platelets count in patients with TTP was 14, range 5–100, compared to 36, range 6–157, in aHUS) and very low ADAMTS13 activity (<10% in all cases). In contrast to patients with aHUS, patients with TTP were less likely to have renal dysfunction, and, in approximately half of cases experiencing acute kidney injury, renal function improved completely after rehabilitation.

The mean age of patients with TTP was 49.4 ± 15.7 years, with women slightly outnumbering men (61.1%) and with comorbidities in 11 cases (61.1%). Hypertension, history of breast cancer, and autoimmune diseases were the most frequently mentioned comorbidities. Patients with COVID-19-associated aHUS were on average 43.2 ± 17.9 years old, slightly predominantly male (57.1%), and mostly had comorbidities. In contrast to patients with the clinical picture of COVID-19-associated TTP, these patients had more or less marked deterioration of renal function, in 32.1% of cases in association with accelerated hypertension. Renal biopsy was performed in 14 patients (50%), most of whom had signs of TMA with predominantly acute features. Although it was difficult to extract data in some cases because precise information was not available, we can conclude that ESRD occurred in 28% of surviving patients with acute renal impairment, whereas renal function was reported as partially or fully recovered in 72% of surviving patients. Compared with patients with TTP, this is a much more pleotropic group, because, in some cases, there were one or more factors or concomitant circumstances that (in addition to COVID-19) alone could trigger TMA, such as diseases related to solid organ transplantation, concurrent treatment with HCQ, pregnancy, chronic infections such as HCV, and several others. However, the most common concomitant disease was arterial hypertension, which was reported in 25% of cases.

Widespread inflammation and endothelial damage are central to the pathogenesis of TMA. However, the extent to which COVID-19 infection is involved in the mechanisms that trigger these events remains to be determined. Patients with TTP may have a congenital deficiency or an acquired reduction in the synthesis of ADAMTS13, leading to platelet aggregation and the formation of intravascular thrombi. We hypothesize that, in COVID-19-associated TTP, insufficient ADAMTS13 activity may be due to antibody synthesis that impairs protease activity as a consequence of COVID-19 infection.

On the other hand, aHUS is associated with endothelial cell damage, either directly (by drugs, infection, or systemic disease) or in the context of defective regulation of the alternative complement pathway, in which case mutations of complement-related proteins are the first hit of the disease. Further endothelial damage (a second hit) can be triggered by several factors, including mild viral upper respiratory tract disease and gastroenteritis [72]. In patients with aHUS, it can be difficult to clearly demonstrate that a trigger exposes latent complement defects. In the COVID-19-associated aHUS group, 50% had laboratory evidence of dysregulation of the alternative complement pathway or already known genetic complement abnormalities. At least in these patients genetically predisposed to aHUS, we speculate that COVID-19 likely represents a second hit by triggering activation of the complement system, whereas, in others, direct damage to the endothelium by COVID-19 may play a role. Moreover, the progression and severity of TMA in patients with COVID-19 could also be due to significant hypercoagulation, which has already been described by many investigators [73]. The pathophysiology of coagulopathy associated with COVID-19 is complex and currently not fully understood. Until recently, it appeared to be related to an enhanced inflammatory response to viral infection rather than to the specific viral properties of SARS-CoV-2 [74]. However, recent research has shown that viral spike protein causes significant ultrastructural changes in whole blood under experimental conditions, leading to extensive spontaneous fibrin network formation and severe impairment of fibrinolysis [75]. Such impairment of clotting in acute COVID-19 infection could contribute significantly to the persistence of microclots in COVID-19 patients, which may also be of major clinical significance for the course and outcome of COVID-19-associated TMA.

Although data on this are currently conflicting, it appears that antiphospholipid antibodies may play a role in COVID-19-induced coagulopathies and possibly also in TMA. These were previously frequently reported in association with COVID-19 infection, but, in the patients presented here, of 12 patients with antiphospholipid antibody analysis performed, only one had anticardiolipin antibodies, and another had beta-2-glycoprotein IgM antibodies. Although APLS cannot be completely ruled out because confirmation of autoantibodies is required at 12 weeks, the likelihood of APLS in these patients appears to be low because the authors did not report a history of autoimmune disease or other evidence of the development of connective tissue disease in other organs. Transient elevations of antibodies to cardiolipin and beta-2-glycoprotein are known to occur in association with various infections [76]. Whether these antibodies may also play a pathophysiological role in the development of COVID-19 related TMA is uncertain at present.

Although COVID-19-associated TMA disorders share many similarities in clinical presentation, the underlying pathophysiology is completely different and requires a specific approach (Figure 1). In TTP, first-line therapy is based on daily TPE delivering deficient ADAMTS13, with or without steroids [77]. The alternative is infusion of FFP, but the results of clinical trials have shown this to be inferior to TPE [78]. In patients with COVID-19-associated TTP, TPE was administered as first-line therapy in 16 of 18 (88.9%) patients, whereas FFP was administered instead in two patients. Corticosteroids were used in 16 of 18 patients (88.9%). Overall, six of eight (75%) patients treated with TPE/FFP and steroids alone recovered. Rituximab, the monoclonal antibody against CD20 on the surface of B cells, was administered to seven of 18 (38.9%) patients, one of whom died. Caplacizumab (inhibitor of VWF-glycoprotein 1b interaction) was administered in two cases of refractory TTP, and both patients recovered. Other immunosuppressive agents (such as vincristine, azathioprine, cyclosphamide, intravenous IgG, and splenectomy) that can be used to treat refractory TTP were not used in patients with COVID-19-associated TTP. A total of 15/28 (53.6%) patients with COVID-19-associated aHUS underwent TPE with or without steroid as initial therapy, resulting in improvement in six patients. A total of 16/28 (57.1%) patients received a C5 complement inhibitor (as monotherapy or after failure of TPE ± steroid to improve the condition), which was prescribed regardless of proven complement system dysfunction or a clear underlying genetic defect. Improvement in renal function occurred in 10 of 16 patients who received complement inhibitor therapy.

In the cohort reported here, three of 18 patients (16.7%) with COVID-19 associated TTP and three of 28 patients (10.7%) with COVID-19 associated aHUS died. In terms of treatment outcomes, mortality in the studied cohort was within the general mortality rate for TTP, which is 10–20% with appropriate treatment [79], but exceeded the most recent data on mortality from aHUS with optimal treatment. Indeed, before anti-complement therapy became available, approximately 50% of aHUS patients developed ESRD with an overall mortality rate of up to 25% [80], but it appears that the mortality rate is currently decreasing steadily. According to a recent meta-analysis, mortality in aHUS was 7.1% between 2005 and 2010, whereas it was 6.1% between 2010 and 2015 [81] and decreased to almost 3% in the era of C5 complement inhibitor use [82]. However, the prognosis of aHUS patients depends on many factors, including concomitant disease, the presence of ESRD, extra-renal involvement, the nature of the complement mutation, and the time interval between the onset of symptoms and the initiation of therapy, including the accessibility of anticomplement therapy [83]; in the case that it is associated with COVID-19, it certainly depends on the severity of the clinical picture of COVID-19. Overall, treatment of COVID-19-related TMA was directed by the underlying pathophysiology of the disease, with the main recognized mechanisms of TMA in patients with COVID 19 being complement-mediated disease and acquired ADAMTS13 deficiency. However, patients with COVID-19 may also develop TMA associated with APLS, Shiga toxin, drugs, or malignancies, and “first hit” complement system abnormalities are thought to be unidentified in many of these patients. The proposed treatment for COVID-19-associated TMA is shown schematically in Figure 1.

## 4. Conclusions

TMA may occur during the course of COVID-19 at the time of viremia or after viral clearance.

We hypothesize that COVID-19 likely represents a second hit of aHUS or TTP that manifests in a certain proportion of genetically predisposed individuals. Early identification of the TMA subtype and appropriate prompt and specific treatment could lead to outcomes comparable to survival and cure statistics for TMA of all causes.

## Figures and Tables

**Figure 1 ijms-23-11307-f001:**
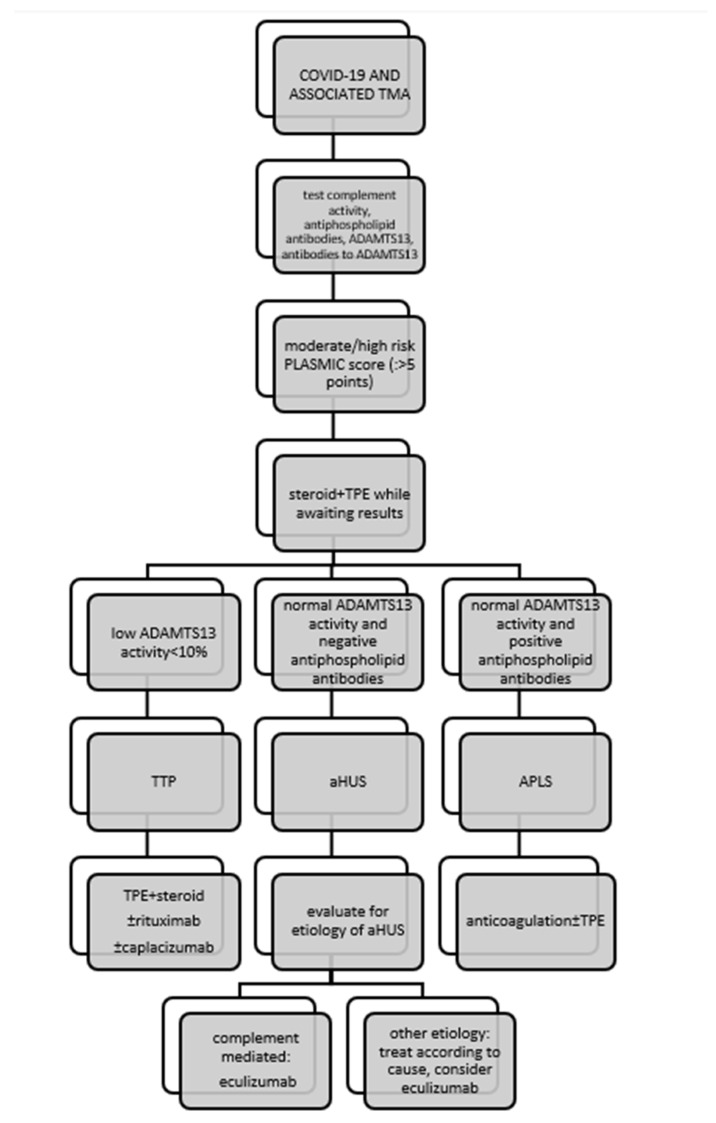
Management of patients with COVID-19-associated thrombotic microangiopathy. Legend: APLS, antiphospholipid syndrome; ADAMTS13, a disintegrin and metalloproteinase with a thrombospondin type 1 motif, member 13; PLASMIC, platelet count, combined hemolysis variable, absence of active cancer, absence of stem-cell or solid-organ transplant, MCV, INR, creatinine.

**Table 1 ijms-23-11307-t001:** Summary of demographic and clinical characteristics of patients presenting as COVID-19-associated TTP.

Reference	Sex	Age	Known Comorbidities	Reported Signs/Symptoms Associated with COVID-19	Proposed Underlying Pathophysiology of TMA (Excluding COVID-19)	Biopsy-Proven TMA	PCR/Serology for SARS-CoV-2	Kidney Function/Serum Creatinine (μmol/L)	Lowest Platelet Count (×10^6^/L)	Accelerated Hypertension	APL Antibodies	TMA-Specific Treatment	Outcome
Kornowski Cohen [29]	F	62	SLE, APLS, CVA, hyperlipidemia trigeminal neuralgia, hypothyroidism	Abdominal pain, mild cough, low-grade fever	ADAMTS13 activity undetectable, borderline anti-ADAMTS13 antibodies	Not performed	+ PCR, serology not done	40 at admission, no further data	41	No	Not assessed (history of APLS)	TPE, steroids, caplacizumab, rituximab	Recovered, no data on renal function
Tehrani [30], Darnahal [31]	F	56	History of locally advanced breast cancer on adjuvant therapy	Fever, dyspnea	ADAMTS13 activity 0.01 IU/mL, + anti-ADAMTS13 antibodies	Not performed	+ PCR, serology not done	No data	41	No	Not assessed	TPE, rituximab	Died—hemorrhagic CVA
Tehrani [30]	F	57	No comorbidities	Severe respiratory symptoms	ADAMTS13 activity 0,86 IU/mL, + anti-ADAMTS13 antibodies	Not performed	+ PCR, serology not done	No data	98	No	Not assessed	Infusion of FFP (shortage of TPE setting), IVIG	Recovered, no data on renal function
Tehrani [30]	M	38	No comorbidities mentioned	No respiratory symptoms, rectorrhagia	ADAMTS13 activity 0,06 IU/mL, + anti-ADAMTS13 antibodies	Not performed	+ PCR, serology not done	No data	5	No	Not assessed	TPE, rituximab	Recovered, no data on renal function
Tehrani [30]	F	25	Third-trimester pregnancy	Fever, dyspnea	ADAMTS13 activity 8%, + anti-ADAMTS13 antibodies	Not performed	+ PCR, serology not done	No data	10	No	Not assessed	TPE, preterm delivery induction	Recovered, no data on renal function
Nicolotti [32]	F	44	Obesity, history of deep vein thrombosis	Weakness, dizziness, abdominal discomfort	ADAMTS13 activity <5%, + anti-ADAMTS13 antibodies	Not performed	+ PCR, serology not done	203 at admission, normal after treatment (no specific data)	7	No	Not assessed	TPE, steroids, rituximab, caplacizumab	Complete recovery of renal function
Albiol [33]	F	57	Arterial hypertension, breast cancer in complete remission	Fever, dry cough, anosmia, dysgeusia	ADAMTS13 activity 2%, + anti-ADAMTS13 antibodies	Not performed	− PCR, + IgG	70 at admission, 53 after treatment	13	No	Not assessed	Steroids, TPE	Complete recovery of renal function
Hindilerden [34]	F	74	Arterial hypertension	Fatigue, cough	ADAMTS13 activity 0.2%, + anti-ADAMTS13 antibodies	Not performed	+ PCR, serology not done	No data	48	No	Not assessed	TPE, steroids	Recovered, no data on renal function
Maharaj [35]	F	69	TTP with multiple relapses, status post PE and cerebral infarct	Acute hypoxic respiratory failure	ADAMTS13 activity 2.9%, + anti-ADAMTS13 antibodies	Not performed	+ PCR, serology not done	No data	100	No	Not assessed	TPE, steroids	Died
Capecchi [36]	F	55	Secondary TTP due to bacterial pneumonia 30 years ago	Mild influenza-like symptoms 1 month prior to current admission	ADAMTS13 activity undetectable, + anti-ADAMTS13 antibodies	Not performed	− PCR, + IgG	281 at admission, 59 at discharge	14	No	Not assessed	TPE, steroids, caplacizumab	Complete recovery of renal function
Dhingra [37]	F	35	No comorbidities	Loose stools 15 days prior to current admission	ADAMTS13 activity undetectable, + anti-ADAMTS13 antibodies	Not performed	+ PCR, serology not done	40–80 (66 at admission, 58 at discharge)	20	No	Negative	TPE, steroids, rituximab	Recovered, intact renal function from the start
Shankar [38]	M	30	No comorbidities, obesity	No respiratory symptoms, back pain, hematuria	ADAMTS13 activity 3%, + anti-ADAMTS13 antibodies	Not performed	+ PCR, serology not done	134 at admission, 68 at discharge	4	No	Not assessed	TPE, steroids, caplacizumab	Complete recovery of renal function
Beaulieu [39]	M	70	Peripheral artery disease, dyslipidemia	Asymptomatic	ADAMTS13 activity <10%, + anti-ADAMTS13 antibodies (low titer)	Not performed	+ PCR, serology not done	106 at admission, 67 at discharge	18	No	Not assessed	Steroids, TPE	Intact renal function, complete neurological recovery
El-Sawalhy [40]	M	49	Arterial hypertension, hyperlipidemia	Left-lower-quadrant abdominal tenderness	ADAMTS13 activity <5%	Not performed	+ (test not specified)	No data	12	No	Not assessed	TPE, steroids	Recovered, no data on renal function
Alhomoud [41]	M	62	Crohn’s disease, G6PD deficiency, TTP	Shortness of breath, generalized weakness, chills	ADAMTS13 activity <5%, − anti-ADAMTS13 antibodies	Not performed	not specified	265 at admission, normal after treatment (no specific data)	21	No	Not assessed	TPE, steroids, rituximab	Complete recovery of renal function
Law [42]	F	47	No comorbidities	Progressive fatigue	ADAMTS13 activity <5%, + anti-ADAMTS13 antibodies	Not performed	+ PCR, serology not done	71 at admission, 56 4 days before discharge	6	No	Negative	TPE, caplacizumab	Complete recovery of renal function
Altowyan [43]	M	39	Diabetes mellitus type 2	Nausea, vomiting, no respiratory symptoms	PLASMIC score 6 (ADAMTS13 activity not assessed)	Not performed	+ PCR, serology not done	77 at admission, no further data	6	No	Negative	TPE, steroids, rituximab	Recovered, renal function intact
Verma [44]	M	21	No comorbidities	Fever, dyspnea, cough	PLASMIC score 7 (ADAMTS13 activity not assessed)	Not performed	+ PCR, serology not done	61 during hospitalization (not further specified)	10	No	Not assessed	FFP (TPE withheld—severe hypotension)	Died (sepsis, progressive TTP)

Legend: APL, antiphospholipid; APLS, antiphospholipid syndrome; COVID-19, coronavirus disease 2019; CVA, cerebrovascular accident; FFP, fresh frozen plasma; G6PD, glucose-6-phosphate dehydrogenase; IVIG, intravenous immunoglobulin; PCR, polymerase chain reaction; SLE, systemic lupus erythematosus; TMA, thrombotic microangiopathy; TPE, therapeutic plasma exchange; TTP, thrombotic thrombocytopenic purpura.

**Table 2 ijms-23-11307-t002:** Summary of demographic and clinical characteristics of patients presenting as COVID-19-associated aHUS.

Reference	Sex	Age	Known Comorbidities	Reported Signs/Symptoms Associated with COVID-19	Proposed Underlying Pathophysiology of TMA (Excluding COVID-19)	Biopsy-Proven TMA	PCR/Serology for SARS-CoV-2	Kidney Function/Serum Creatinine (μmol/L)	Lowest Platelet Count (×10^6^/L)	Accelerated Hypertension	APL Antibodies	TMA-Specific Treatment	Outcome
Ville [45]	F	28	Hereditary aHUS, CKD 3, arterial hypertension	No respiratory symptoms, fever, dysphagia, headache	Heterozygous pathogenic variant in the membrane cofactor protein-encoding gene	Not performed	+ PCR, serology not performed	230 at admission, peaked at 256, 194 at discharge, 176 on follow-up after 1 month	106	No	Not assessed	Eculizumab	Partial recovery of renal function
El Sissy [46]	M	66	Not mentioned	Mild respiratory symptoms	Pathogenic variant of CFH gene, reduced factor H plasma level, mildly increased sC5b-9	Yes: glomerular + arteriolar thrombi, EC detachment	+ PCR, serology not performed	884 at admission—HD	50	Yes	Not assessed	None	Renal failure
El Sissy [46]	M	71	ESRD due to nephroangiosclerosis, kidney transplant recipient	Moderate respiratory symptoms (low grade oxygen therapy)	Pathogenic variant of C3 gene, very high trough levels of CNI (transplanted kidney)	Yes: performed 3 weeks after TMA resolution; GBM duplication, glomerulosclerosis	+ PCR, serology not performed	203 at admission, 150 (baseline value) after 1 month	16	No	Not assessed	TPE, eculizumab	Recovery of renal function to baseline
El Sissy [46]	M	35	Not mentioned	Mild respiratory symptoms	Pathogenic variant of CFI gene, reduced factor I plasma level, + factor H autoantibodies	Yes: glomerular + arteriolar thrombi	+ PCR, serology not performed	699 at admission—HD	11	Yes	Not assessed	TPE, eculizumab	Renal failure
El Sissy [46]	F	26	ESRD due to focal segmental glomerulosclerosis, kidney transplant recipient	Mild respiratory symptoms	Risk haplotype in the membrane-cofactor protein gene, + factor H autoantibodies, CNI (transplanted kidney)	Not performed	+ PCR, serology not performed	637 at admission, 371 after 4 months	22	Yes	Not assessed	TPE, eculizumab, rituximab	Partial recovery of renal function
Mat [47]	M	39	IgA nephropathy, CKD 3, hypertension, esophagitis	No respiratory symptoms, diarrhea, fever	C3 gene mutation	Yes: GBM duplication, mucoid thickening + obliteration of the lumen of a small artery	Both +	416 at admission, became HD-dependent during hospitalization	80	No	Negative	TPE, steroids, eculizumab	Renal failure
Kurian [48]	F	31	No comorbidities	No respiratory symptoms, headache, nausea, diarrhea	Low C4, + factor H autoantibodies	Not performed	+ PCR, serology not performed	540 at admission, HD-dependent during hospitalization, 97 on follow-up	25	No	Not assessed	TPE, steroids, eculizumab	Partial recovery of renal function
Kurian [48]	M	25	Previous episode of TMA following influenza A	No respiratory symptoms, nausea, malaise, fever	Decreased C3, normal C4	Not performed	+ PCR, serology not performed	219 at admission, peaked at 653 (no HD), 89 2 weeks after discharge	32	No	Not assessed	TPE, eculizumab	Complete recovery of renal function
Korotchaeva [49]	F	49	Not mentioned	No respiratory symptoms, abdominal pain, diarrhea, vomiting, fever	Decreased C3, increased C5b-9	Yes: mucoid swelling of arterioles, glomerular capillary loops ischemia	+ PCR, serology not performed	210 at admission, peaked at 558—HD, 300 after 4 months	157	No	Negative	Eculizumab	Partial recovery of renal function
Pinte [50]	M	23	Arterial hypertension, previous episodes of accelerated hypertension	No respiratory symptoms, headache	Heterozygous variant with unknown significance in CFI, risk polymorphisms in CFH	Yes: glomerular capillary lumina occluded by fibrin thrombi, severely narrowed or occluded arterial lumina	+ PCR, serology not performed	424 at admission, peaked at 946—HD	125	Yes	+ lupus anticoagulant	None (late diagnosis, lack of availability of eculizumab)	Renal failure
Utebay [51]	M	76	Atrial fibrillation	Dyspnea, cough, cardiogenic shock	Severely decreased complement levels	Not performed	+ PCR, serology not performed	88 at admission, peaked at 357—RRT, 170 at discharge	12	No	Negative	TPE, steroids, eculizumab	Partial recovery of renal function
Logan [52]	M	40	No comorbidities	Weakness, fever, altered mental status	Dysregulation of the alternative pathway of the complement	Not performed	+ (test not specified)	714 at admission—HD, improvement during hospitalization (not further specified)	7	No	Not assessed	TPE	Recovery of renal function
Jhaveri [53]	F	69	Asthma	Cough, fever, dyspnea	Low factor H, HCQ	Yes: widespread glomerular thrombi	Both +	64 at admission, required HD later on	14	No	Elevated β-2-GPI IgM, other negative	Eculizumab	Died
El Sissy [46]	F	38	ESRD due to IgA nephropathy, kidney transplant recipient	Mild respiratory symptoms	CNI (transplanted kidney)	Yes: extensive EC detachment from GBM	+ PCR, serology not performed	283 at admission, 159 (baseline value) 6 months after treatment	103	No	Not assessed	None (decrease in tacrolimus dosage)	Recovery of renal function to baseline
Bascunana [54]	M	40	ESRD due to Liddle syndrome, kidney transplant recipient	Fever, dyspnea, diarrhea, abdominal pain	HCQ, CNI (transplanted kidney)	Not performed	+ PCR, serology not performed	326 at admission, peaked at 777—HD, 168 at discharge	12	No	Negative	TPE	Recovery of kidney function
Jespersen Nizamic [55]	F	49	ESRD due to FSGS, kidney transplant recipient	No respiratory symptoms, diarrhea, menorrhagia, acute pancreatitis	CNI (transplanted kidney)	Yes: significant narrowing/obliteration of the hilar arteriolar lumina, intimal edema, fragmented RBC and fibrin thrombi	+ PCR, serology not performed	357 at admission, peaked at 539, 151 10 days after discharge	52	No	Not assessed	Reduced dose of CNI	Complete recovery of kidney function
Sharma [56]	F	63	Metastatic cholangiocarcinoma	Dyspnea	Metastatic carcinoma, gemcitabine	Yes: Acute features of TMA–not further specified	Not specified	Approximately 100 at admission, peaked at approximately 640—HD	53	No	Not assessed	TPE, steroids	Palliative care, no data on kidney function
Tarasewicz [57]	M	41	ESRD due to IgA nephropathy, kidney transplant recipient	Diarrhea, hypertension; presented with pneumonia 1 month earlier	CNI (transplanted kidney)	Yes: fragmented RBCs within edematous glomerular capillary walls	+ PCR 1 month prior	325 at admission, started HD 4 months later	115	Yes	Not assessed	TPE	Graft failure
Safak [58]	M	34	Arterial hypertension	Blurred vision	Severe hypertension (after normalization of hypertension, TMA did not resolve)	Not performed	+ PCR, serology not performed	433 at admission, 106 at follow-up	28	Yes	Negative	Eculizumab	Complete recovery of renal function
Gill [59]	M	32	Childhood leukemia, orthotopic heart transplant, CKD 3	Fever, cough, dyspnea, chest pain	Sirolimus (transplanted heart)	Yes: fibrin thrombi in arterioles without endocapillary hypercellularity or inflammation	Not specified	685 at admission, 424 7 days after eculizumab, 202 at discharge	43	No	Negative	TPE, eculizumab, ravulizumab	Complete recovery of renal function
Aminimoghaddam [60]	F	21	Drug addict (amphetamine)	Fever, dry cough	Pregnancy, HCQ	Not performed	+ PCR, serology not performed	495 at admission, 80 at discharge	20	No	Negative	TPE	Complete recovery of renal function, stillbirth
Boudhabhay [61]	M	46	Arterial hypertension, obesity	No COVID-19 associated symptoms	Elevated C5b9, low C4, genetic testing insignificant	Yes: glomerular fibrin thrombi, myxoid intimal alterations of arterioles and arteries	− PCR, + IgG	169 at admission, peaked at 660-RRT, 109 at discharge	90	Yes	Negative	Eculizumab	Complete recovery of renal function
Dorooshi [62]	F	81	Arterial hypertension	Fever, dry cough, dyspnea, anorexia, vomiting	Not known	Not performed	+ PCR, serology not performed	805 at admission	52	No	Not assessed	None	Died before treatment
Airoldi [63]	M	56	Liver cirrhosis (Child–Pugh B), HCV	Bilateral pneumonia	HCV, fondaparinux	Not performed	+ PCR; serology not performed	No data	36	No	Not assessed	Steroids	Recovered, no data on kidney function
Korotchaeva [49]	M	39	Not mentioned	Fever, dry cough, malaise, dyspnea, anosmia	Not known	Not performed	+ PCR, serology not performed	1035 at admission—HD	51	Yes	Negative	TPE, steroids, eculizumab	Renal failure
Korotchaeva [49]	F	66	Diabetes mellitus type 2	ARDS	Not known	Not performed	+ PCR, serology not performed	70 at admission, peaked at 193	6	No	Negative	FFP transfusions, tocilizumab, eculizumab	Died
Elkayam [64]	F	44	No comorbidities	Fever, dry cough, dyspnea	Not known	Not performed	+ PCR, serology not performed	194 on day 2, worsened to HD dependency (not further specified)	18	No	Not assessed	TPE, FFP	Partial recovery of renal function
Gandhi [65]	M	27	Arterial hypertension, CKD	Cough, lethargy, effort intolerance, loss of vision, jejunal perforation	Not known	Yes: ischemic glomerular injury, fibrin deposition and fragmented RBCs in interlobar arterial wall	+ PCR, serology not performed	1963 at admission—HD	Lack of data	Yes	Not assessed	High-dose methylprednisolone, bevacizumab, eculizumab	Renal failure

Legend: aHUS, atypical hemolytic uremic syndrome; APL, antiphospholipid; CKD, chronic kidney disease; CFH, complement factor H; CFI, complement factor I; CNI, calcineurin inhibitor; COVID-19, coronavirus disease 2019; EC, endothelial cells; ESRD, end-stage renal disease; FFP, fresh frozen plasma; GBM, glomerular basement membrane; GPI, glycoprotein; HCV, hepatitis C; HCQ, hydroxychloroquine; HD, hemodialysis; PCR, polymerase chain reaction; RRT, renal replacement therapy; TMA, thrombotic microangiopathy; TPE, therapeutic plasma exchange.

**Table 3 ijms-23-11307-t003:** Outcomes in patients with COVID-19-associated TMA in relation to received therapy.

Patients with TTP	FFP	TPE + Steroid	TPE ± Steroid + Caplacizumab	TPE ± Steroid + Rituximab	TPE + Steroid + Rituximab + Caplacizumab
** *N* ** ** = 18**	2	5 + 1 *	3	5	2
**Recovered**	1	5	3	4	2
**Died**	1	1		1	
**Patients with aHUS**	TPE or FFP only	Steroid only	TPE + steroid only	Eculizumab or ravolizumab	No specific therapy or else
** *N* ** **= 28**	5	1	1	16 * 12 also TPE ± steroid	5
**Recovery of renal function**					
**Complete**	1			5	2
**Partial/unknown magnitude**	3	1	1	5	
**ESRD**	1			4	2
**Died**				2	1

* Preterm delivery induced, did not receive steroid. Legend: aHUS, atypical hemolytic uremic syndrome; ESRD, end-stage renal disease; FFP, fresh frozen plasma; TPE, therapeutic plasma exchange; TTP, thrombotic thrombocytopenic purpura.

## Data Availability

Not applicable.

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
