# Peer review of "Coronavirus Disease 2019-Associated Thrombotic Microangiopathy: Literature Review"

_ijms, 2022, doi:10.3390/ijms231911307_

Round 1

Reviewer 1 Report

This manuscript reviews the literature of “TMA” among patients with COVID-19 infection. Overall, 46 cases were identified, including 18 cases of “thrombotic thrombocytopenic purpura (TTP) or probably TTP” and 28 cases of “atypical hemolytic uremic syndrome (aHUS)”. Overall, the disease classification scheme is inconsistent with the current state of knowledge. It is confusing and incoherent. This misclassification affects the presentation of the literature and obscures the lessons to be learned. Only three examples will be mentioned below.

Firstly, the authors state “TMA syndromes share certain pathologic and clinical features. The most important pathologic feature is widespread microvascular injury manifested by thrombi of platelets and fibrin in capillaries and arterioles with characteristic abnormalities of the endothelium and vessel wall. Clinical features include microangiopathic hemolytic anemia (MAHA), thrombocytopenia, and organ damage (AKI, neurologic abnormalities)[6].” (Lines 44-49)

They then classify TMA in three groups: thrombotic thrombocytopenic purpura (TTP), STEC-HUS and aHUS (lines 61-66), failing to recognize that microvascular injury and fibrin thrombi are not features of TTP and that platelet thrombi are not always detectable in all “aHUS” cases. Their scheme also ignores that fact in the group of what they classify as “aHUS” the pathology can be quite different and some patients do not have kidney injury or neurologic abnormalities.  

Secondly, in lines 74-76, the authors state that “aHUS, ……, is associated with uncontrolled activation of the alternative complement pathway.” This is inconsistent with their classification of “aHUS” and what they depict in Figure 2, in which they classify aHUS in two categories: complement mediated, to be treated with eculizumab, and other etiology, to be treated according to cause.

Thirdly, the authors claim that “…..,TPE has been shown to be beneficial in the treatment of almost all types of TMA 142 (not just TTP).” (Lines 141-143) This claim is not supported with any evidence in the literature.

Author Response

Dear reviewers,

thank you for reviewing the article and all your valuable suggestions, which the authors have tried to take into account as much as possible, considering the comments of all 3 reviewers, which are somewhat contradictory in some places. We ask for your understanding in these cases, as we have tried to please everyone and correct our article to the best of our ability.

Below, we present point-by-point responses to your comments.

Best regards,

Željka Večerić-Haler, corresponding author

Review1

This manuscript reviews the literature of “TMA” among patients with COVID-19 infection. Overall, 46 cases were identified, including 18 cases of “thrombotic thrombocytopenic purpura (TTP) or probably TTP” and 28 cases of “atypical hemolytic uremic syndrome (aHUS)”. Overall, the disease classification scheme is inconsistent with the current state of knowledge. It is confusing and incoherent. This misclassification affects the presentation of the literature and obscures the lessons to be learned. Only three examples will be mentioned below.

The classification of TMAs in the current literature is often ambiguous and somewhat confusing. The challenge regarding this semantic dilemma arises from historical aspects and the evolving understanding of the pathophysiological basis of the various TMA entities. Different clinical viewpoints from various medical specialties, including paediatricians, haematologists, immunologists, and nephrologists, contribute to the pertinent differences in the use of TMA nomenclature. For our study, we used the classification of Goodship TH et al (Kidney Int. 2017 Mar;91(3):539-551. doi: 10.1016/j.kint.2016.10.005).This classification is closest to us nephrologists, as it also deals with other complement-related renal diseases. It also shows a high degree of similarity to the recently updated classification by Aigner at al (Aigner C, et al. An updated classification of thrombotic microangiopathies and treatment of complement gene variant-mediated thrombotic microangiopathy, Clinical Kidney Journal, Volume 12 , Issue 3, June 2019, Pages 333-337).

Of course, we must admit that none of the classifications presented so far is perfect, most likely due to the still existing misunderstanding of the etiopathogenesis and pathophysiology. Therefore, in case of choosing a different classification and depending on the specialty of the reviewer, we might encounter similar criticism. However, we are aware that we have created inconsistency and confusion by using terms such as "probable." Therefore, and also in agreement with the opinion of another reviewer, we have corrected the terminology accordingly so that we have strictly followed the terminology of the classification chosen for this study.

Because this manuscript is not intended to be an overview of TMA classification and also in agreement with reviewer 3's comment, we have shortened and adapted the introductory section accordingly (including the classification of TMAs and Figure 1, which has been removed from the revised version of the manuscript)

Firstly, the authors state “TMA syndromes share certain pathologic and clinical features. The most important pathologic feature is widespread microvascular injury manifested by thrombi of platelets and fibrin in capillaries and arterioles with characteristic abnormalities of the endothelium and vessel wall. Clinical features include microangiopathic hemolytic anemia (MAHA), thrombocytopenia, and organ damage (AKI, neurologic abnormalities)[6].” (Lines 44-49)

They then classify TMA in three groups: thrombotic thrombocytopenic purpura (TTP), STEC-HUS and aHUS (lines 61-66), failing to recognize that microvascular injury and fibrin thrombi are not features of TTP and that platelet thrombi are not always detectable in all “aHUS” cases. Their scheme also ignores that fact in the group of what they classify as “aHUS” the pathology can be quite different and some patients do not have kidney injury or neurologic abnormalities.  

We agree with the reviewer's comment on the pathological features of TMA syndromes. We have corrected the text accordingly. We also agree that some patients do not have kidney injury or neurologic abnormalities. For this study, an electronic search of the PubMed database was performed using keywords and their combinations: "thrombotic microangiopathy, COVID-19, thrombotic thrombocytopenic purpura, atypical hemolytic uremic syndrome." The selected keywords and their combinations do not exclude patients who do not present with kidney injury or neurologic abnormalities, but such patients are very likely to be more difficult to recognize and diagnose and for this reason are virtually unrepresented in the context of COVID-19-related TMA.

Secondly, in lines 74-76, the authors state that “aHUS, ……, is associated with uncontrolled activation of the alternative complement pathway.” This is inconsistent with their classification of “aHUS” and what they depict in Figure 2, in which they classify aHUS in two categories: complement mediated, to be treated with eculizumab, and other etiology, to be treated according to cause.

We stated in our manuscript that "The classification into primary and acquired TMA is not absolute, as hereditary TMA requires a trigger factor and acquired TMA may also have a genetic background...". This statement is in complete agreement with the classification used. However, we agree with the reviewer's critique of Figure 2, which represents our proposed treatment for a patient with COVID 19-related TMA. In aHUS that has a recognized cause (e.g., medications, ...), it is necessary to eliminate the cause when/if possible. According to our current understanding of aHUS and its association with complement abnormalities (based on the literature and our own experience), it is perfectly reasonable to reconsider anti-complement therapy if we have been unsuccessful with removal of the recognized cause/trigger alone, regardless of unproven functional or genetic complement abnormalities, and if there are no contraindications on the part of the patient. We have adapted Figure 2 (Figure 1 in the new version of the manuscript) accordingly in light of this thought.

Thirdly, the authors claim that “…..,TPE has been shown to be beneficial in the treatment of almost all types of TMA 142 (not just TTP).” (Lines 141-143) This claim is not supported with any evidence in the literature.

In most cases of both diseases, immediate treatment is given with plasma exchange initially and monoclonal therapy (rituximab in TTP and eculizumab in aHUS) as the mainstay of therapy. Appropriate references are added.

Reviewer 2 Report

This is a very well written review that is missing some very important key elements with regards to COVID-19 and kidney injury.

The key take away conclusion from this study is that COVID-19 likely represents a second hit of aHUS or TPP in genetically predisposed individuals.  This reviewer only partially accepts this conclusion in that it likely applies to a subset of COVID-19 patients but not all of the patients who develop kidney injury during COVID-19 or shortly afterwards.  This reviewer kindly requests the authors consider a slight modification to their conclusion.

COVID-19 by itself can result in significant kidney injury in hospitalized patients.  Additional references on this topic would strengthen this manuscript.  Some authors have proposed that COVID-19 is inducing vasoconstrictions associated with myocarditis, pericarditis, etc.  This has parallels for this article.

COVID-19 is associated with microthrombi.  This article should overview this as it is highly relevant to this article.  This article is highly relevant and should be considered by the authors: doi: 10.1042/BSR20210611

Kaneko et al. describe dysregulation of humoral immune responses by COVID-19: doi: 10.1016/j.cell.2020.08.025. This should be considered in the context of this article and viral induced autoantibodies.

Dr. Tomera is treating COVID-19 patients with high dose famotidine and high dose celecoxib.  See: https://papers.ssrn.com/sol3/papers.cfm?abstract_id=3646583. None of his patients develop kidney injuries.  This article appears to be using the old tried and tested solutions for treating viral injured patients.  Relevant articles indicating reductions of kidney injuries for COVID-19 patients associated with candidate treatments would strength this article.  

Author Response

Dear reviewers,

thank you for reviewing the article and all your valuable suggestions, which the authors have tried to take into account as much as possible, considering the comments of all 3 reviewers, which are somewhat contradictory in some places. We ask for your understanding in these cases, as we have tried to please everyone and correct our article to the best of our ability.

Below, we present point-by-point responses to your comments.

Best regards,

Željka Večerić-Haler, corresponding author

Review 2

This is a very well written review that is missing some very important key elements with regards to COVID-19 and kidney injury.

The key take away conclusion from this study is that COVID-19 likely represents a second hit of aHUS or TPP in genetically predisposed individuals.  This reviewer only partially accepts this conclusion in that it likely applies to a subset of COVID-19 patients but not all of the patients who develop kidney injury during COVID-19 or shortly afterwards.  This reviewer kindly requests the authors consider a slight modification to their conclusion.

We thank reviewer for this comment. The conclusion was appropriatelly modified.

COVID-19 by itself can result in significant kidney injury in hospitalized patients.  Additional references on this topic would strengthen this manuscript.  Some authors have proposed that COVID-19 is inducing vasoconstrictions associated with myocarditis, pericarditis, etc.  This has parallels for this article.

COVID-19 is associated with microthrombi.  This article should overview this as it is highly relevant to this article.  This article is highly relevant and should be considered by the authors: doi: 10.1042/BSR20210611

Kaneko et al. describe dysregulation of humoral immune responses by COVID-19: doi: 10.1016/j.cell.2020.08.025. This should be considered in the context of this article and viral induced autoantibodies.

We thank the reviewer for this comment. The systemic and renal injury associated with COVID 19 is undoubtedly broad, and we mentioned it briefly in the introductory section, which is now more concise. The request of another reviewer was to shorten the introduction significantly and focus it only on the topic under discussion. In what follows, therefore, we wanted to focus on the main topic of this article, namely TMA in the context of COVID 19, so that the broader spectrum of systemic/immunologic and renal pathology associated with COVID -19 is not discussed further.

Dr. Tomera is treating COVID-19 patients with high dose famotidine and high dose celecoxib.  See: https://papers.ssrn.com/sol3/papers.cfm?abstract_id=3646583. None of his patients develop kidney injuries.  This article appears to be using the old tried and tested solutions for treating viral injured patients.  Relevant articles indicating reductions of kidney injuries for COVID-19 patients associated with candidate treatments would strength this article.  

Our review article addresses only the cases and case series of TMA associated with COVID- 19 described through February 22 and not cases of other COVID-19-associated pathologies. It also discusses the treatments used in the published TMA cases and draws conclusions based solely on these publications. Discussion of other possible treatment options would be interesting but would completely detract from the focus of our paper.

Reviewer 3 Report

Authors provided an interesting review on all published cases of TMA in association to COVID-19.

Introduction (part 1) is definitively too long and not focused on the topic, which is COVID-19 infection as cause/trigger of thrombotic microangiopathy (TMA).

You should mention that it is increasingly recognized that TMA may present with hypertension and renal function impairment (in this case is essential to prove TMA diagnosis with renal biopsy)even with no or mild thrombocytopenia and microangiopathic hemolytic anemia. You con choose one of the following references to support this point: Ardissino G, Eur J Intern Med 2013; De Serres SA, NDT 2009; Sallée M, BMC Nephrol 2013.

Table 1 should be removed because it’s not a review of TMA classification

The following sentence is not correct:

“Certain cases[25-27] were excluded due to missing data or to the fact that the presented case also received vaccine against COVID-19. We also excluded a case of probable TTP due to lack of data and the possibility of tocilizumab-associated TTP[28].”

In fact, vaccine against covid and tocilizumab (tocilizumab-related TMA has never been described) are not reasonable exclusion criteria. Consequently these patients should be considered to be included.

The following sentence is unacceptable:

“... reported as acquired TTP or probable acquired TTP (in cases assessed by PLASMIC score since measurement of ADAMTS13 activity was not available) (Table 1), and a total of 28 cases presented as mixed forms of TMA associated with multifactorial triggers but without proven abnormalities of ADAMTS13 activity or proven ADAMTS13 inhibitors or proven Shiga toxin (Table 2). In some of these patients, the definition of TMA subtype was not entirely reliable because of a lack of information, but the clinical presentation, laboratory or other findings, ADAMTS13 activity or response to more differentiating treatment, such as eculizumab, were more suggestive of aHUS. In this case, we classified the patients as »probable aHUS.”

The classification of all cases is not correct.

You shouls speak about TTP in case of ADAMTS13 activity <10%, about aHUS in case of genetic causative mutations in complement factors, and about TMAs (as a generic term) in case of non-TTP and non-aHUS cases of TMA.

Consequently you should re-classify the entire cohort.

And Figure 2 should be changed, accordingly.

Different scoring systems (like ‘PLASMIC score,’ ‘French score,’ and ‘Bentley score’) predict severe ADAMTS13 deficiency (and consequently TTP diagnosis) with high sensitivities and specificities. Based on the above scoring systems, Authors could suppose if the probability of TTP is low in a patient but can’t do a diagnosis.

Moreover, why did you choose PLASMIC score and not the others?

Re-evaluation of all the identified variants was based on ClinVar. Variants were reported with the use of specific standard terminology - “pathogenic” , “likely pathogenic”, “uncertain significance” , “likely benign” and “benign”- according to the ACMG guidelines (Richards S, et al. Genetics in medicine: official journal of the American College of Medical Genetics 2015).

In Table 1 and Table 2. Clinical signs and symptoms associated with COVID should also include information on kidney function during hospitalization

Line 238 and 241. Authors should specify the meaning of “showed laboratory evidence of dysregulation of the alternative complement pathway”. Only C3 reduction, AP50 or Wieslab test?

Authors claim that “Renal biopsy was performed in 14 patients, all of whom had patohistological signs of TMA.” TMA can histopathologically present in different ways, and Authors should schematically report them in COVID-related TMA.

The review of the literature is quite incomplete.

For example, the following articles on cases affected by covid-19-related TMA should be considered to be included:

—Menter T et al. Postmortem examination of COVID-19 patients reveals diffuse alveolar damage with severe capillary congestion and variegated findings in lungs and other organs suggesting vascular dysfunction. Histopathology. 2020

—Korotchaeva J et al. Thrombotic Microangiopathy Triggered by COVID-19: Case Reports. Nephron 2022;146:197–202

—Tahir Dalkıran et al. Thrombotic Microangiopathy in a Severe Pediatric Case of COVID-19. Clinical Medicine Insights: Pediatrics 2021

Authors should perform a deeper review of the published literature on TMA in patients with COVID-19 through PubMed/MEDLINE and Google Scholar databases.

Author Response

Dear reviewers,

thank you for reviewing the article and all your valuable suggestions, which the authors have tried to take into account as much as possible, considering the comments of all 3 reviewers, which are somewhat contradictory in some places. We ask for your understanding in these cases, as we have tried to please everyone and correct our article to the best of our ability.

Below, we present point-by-point responses to your comments.

Best regards,

Željka Večerić-Haler, corresponding author

Review 3

Authors provided an interesting review on all published cases of TMA in association to COVID-19.

Introduction (part 1) is definitively too long and not focused on the topic, which is COVID-19 infection as cause/trigger of thrombotic microangiopathy (TMA).

We thank the reviewer for his comment. In accordance with this comment, we have adjusted and shortened the introductory section to increase focus.

You should mention that it is increasingly recognized that TMA may present with hypertension and renal function impairment (in this case is essential to prove TMA diagnosis with renal biopsy) even with no or mild thrombocytopenia and microangiopathic hemolytic anemia. You con choose one of the following references to support this point: Ardissino G, Eur J Intern Med 2013; De Serres SA, NDT 2009; Sallée M, BMC Nephrol 2013.

We improved the manuscript by adding suggested information and references in the introductory section.

Table 1 should be removed because it’s not a review of TMA classification

We agree that our manuscript is not intended to be an overview of the TMA classification. Therefore, we have removed Figure 1.

The following sentence is not correct:

“Certain cases[25-27] were excluded due to missing data or to the fact that the presented case also received vaccine against COVID-19. We also excluded a case of probable TTP due to lack of data and the possibility of tocilizumab-associated TTP[28].”

In fact, vaccine against covid and tocilizumab (tocilizumab-related TMA has never been described) are not reasonable exclusion criteria. Consequently these patients should be considered to be included.

Considering that there are already cases of TMA associated with the COVID-19 vaccine, and that we ourselves already have several cases of such patients (publication in preparation), we excluded patients with TMA possibily induced by vaccination. We also excluded all patients in whom there were insufficient data to unequivocally confirm the author's clinical suspicion of one of the forms of TMA, because inclusion of such vague cases could be misleading. Tocilizumab was used extensively during the covid epidemic. TMA induced by tocilizumab has been described (Ito S, et al. Trombotic microangiopathy developing subsequent to tocilizumab therapy in a patient with TAFRO syndrome. Rinsho Ketsueki. 2018;59(11):2432-2437. Japanese. doi: 10.11406/rinketsu.59.2432. PMID: 30531139;Jewell P, et al. Tocilizumab-associated multifocal cerebral thrombotic microangiopathy. Neurol Clin Pract. 2016 Jun;6(3):e24-e26. doi: 10.1212/CPJ.0000000000000220. PMID: 27347443; PMCID: PMC4909526).

However, because this case was excluded primarily because of missing data, we excluded the statement related to tocilizumab to avoid confusion.

The following sentence is unacceptable:

“... reported as acquired TTP or probable acquired TTP (in cases assessed by PLASMIC score since measurement of ADAMTS13 activity was not available) (Table 1), and a total of 28 cases presented as mixed forms of TMA associated with multifactorial triggers but without proven abnormalities of ADAMTS13 activity or proven ADAMTS13 inhibitors or proven Shiga toxin (Table 2). In some of these patients, the definition of TMA subtype was not entirely reliable because of a lack of information, but the clinical presentation, laboratory or other findings, ADAMTS13 activity or response to more differentiating treatment, such as eculizumab, were more suggestive of aHUS. In this case, we classified the patients as »probable aHUS.”

The classification of all cases is not correct.

You shouls speak about TTP in case of ADAMTS13 activity <10%, about aHUS in case of genetic causative mutations in complement factors, and about TMAs (as a generic term) in case of non-TTP and non-aHUS cases of TMA.

Consequently you should re-classify the entire cohort.

And Figure 2 should be changed, accordingly.

We thank the reviewer for this valid criticism. We agree with the reviewer that the paragraph mentioned is inadequate, including inconsistency in classification of patients. We followed the recommendation and reclassified the patients very strictly and exclusively according to the classification chosen for our study and changed the controversial terminology. The reviewer is asked to consider the fact that the analysis of the present study is based on the classification of Goodship et al. According to the chosen classification, TMA is divided into TTP and HUS. The latter classification further distinguishes into STEC HUS and aHUS. In aHUS, a distinction is made between primary aHUS (detected genetic mutations at the complement level) and secondary aHUS (all other forms).

In the article, we have changed and marked the information in all relevant places.

Round 2

Reviewer 1 Report

In this revised manuscript, the authors have made revisions to address some of the concerns raised in the original review. Nevertheless the presentation of disease classification and its related discussions remain incoherent and confusing. The text reflects lack of practical experience and critical knowledge of the relevant literature.  Without a rational, fact-based scheme of disease classification, the authors are unable to provide an in formative review of the relevant literature. For a better appreciation of the complexity of the subject and an informed approach to their review, the authors may consult doi: 10.1016/s0046-8177(88)80093-5, doi: 10.1007/s004670050337, doi: 10.1007/s00467-003-1385-9, doi: 10.1016/0049-3848(85)90180-x and Wintrobe's Clinical Hematology 14/e Chapter 49

Author Response

Dear reviewers,

thank you for the detailed review of our article and the given suggestions. We have done our best to take into account your comments and suggestions, which we believe have helped to improve the article.

Željka Večerić-Haler, the corresponding author

Reviewer 1

In this revised manuscript, the authors have made revisions to address some of the concerns raised in the original review. Nevertheless the presentation of disease classification and its related discussions remain incoherent and confusing. The text reflects lack of practical experience and critical knowledge of the relevant literature.  Without a rational, fact-based scheme of disease classification, the authors are unable to provide an in formative review of the relevant literature. For a better appreciation of the complexity of the subject and an informed approach to their review, the authors may consult doi: 10.1016/s0046-8177(88)80093-5, doi: 10.1007/s004670050337, doi: 10.1007/s00467-003-1385-9, doi: 10.1016/0049-3848(85)90180-x and Wintrobe's Clinical Hematology 14/e Chapter 49.

In the first round of the review, we explained why we chose the Goodship et al classification for this review of TMA associated with Covid-19. We attempted to consistently place patient reports into the appropriate category of the chosen classification. We also emphasized that our article is not an article about the TMA classification but uses a selected valid classification to place patients into the appropriate categories. We emphasize here that the patients were not diagnosed by us, the authors of the present article, but that we simply reviewed the data reported in the covered publications and the diagnoses made by the authors of the published articles and ensured that only these met the required criteria for a diagnosis. Articles with missing key data were excluded to avoid misleading information.

The reviewer in round 2 criticized our lack of practical experience and critical knowledge of the relevant literature in this field. Unfortunately, we do not understand this criticism, nor do we know which parts of the article the criticism refers to. The reviewer suggests the literature listed below, which we do not consider relevant to the point of the article, with the exception of the article by Nester et al, which we are completely familiar with, but which is a review paper that is older and does not yet take into account all the new aspects and findings about aHUS, especially the involvement of the complement system in secondary forms of the disease. In fact, it was this group of experts that later revised and renewed the classification of TMA that was chosen as the matrix for our presentation and cited in the references!

We are sorry, but on the basis of the above arguments, we cannot further correct the article in line with the reviewer's suggestions.

The suggested literature by the reviewer 1:

Mayer CL, Leibowitz CS, Kurosawa S, Stearns-Kurosawa DJ. Shiga toxins and the pathophysiology of hemolytic uremic syndrome in humans and animals. Toxins (Basel). 2012 Nov 8;4(11):1261-87. doi: 10.3390/toxins4111261. PMID: 23202315; PMCID: PMC3509707.

Taylor CM, Chua C, Howie AJ, Risdon RA; British Association for Paediatric Nephrology. Clinico-pathological findings in diarrhoea-negative haemolytic uraemic syndrome. Pediatr Nephrol. 2004 Apr;19(4):419-25. doi: 10.1007/s00467-003-1385-9. Epub 2004 Feb 24. PMID: 14986082.

Asada Y, Sumiyoshi A, Hayashi T, Suzumiya J, Kaketani K. Immunohistochemistry of vascular lesion in thrombotic thrombocytopenic purpura, with special reference to factor VIII related antigen. Thromb Res. 1985 Jun 1;38(5):469-79. doi: 10.1016/0049-3848(85)90180-x. PMID: 2861671.

Nester CM, Barbour T, de Cordoba SR, Dragon-Durey MA, Fremeaux-Bacchi V, Goodship TH, Kavanagh D, Noris M, Pickering M, Sanchez-Corral P, Skerka C, Zipfel P, Smith RJ. Atypical aHUS: State of the art. Mol Immunol. 2015 Sep;67(1):31-42. doi: 10.1016/j.molimm.2015.03.246. Epub 2015 Apr 3. PMID: 25843230.

Reviewer 2 Report

The authors should review and consider citing: doi: 10.1042/BSR20210611 

The text and layout of Figure 1 may not be as intended.  Please review and improve for publication.  The text for the second box from the top does not fit the surrounding shape.

The conclusion includes: "COVID-19 likely represents a second hit of aHUS or TTP that manifests in certain part of genetically predisposed individuals.".  Please consider restating this as a hypothesis.  Likewise, please adjust similar statements in the article.  The article indicates candidate genes that can be examined in patients to evaluate this hypothesis; the text can be update to indicate evaluation options.

Author Response

Dear reviewers,

thank you for the detailed review of our article and the given suggestions. We have done our best to take into account your comments and suggestions, which we believe have helped to improve the article.

Željka Večerić-Haler, the corresponding author

Reviewer 2

The authors should review and consider citing: doi: 10.1042/BSR20210611 

We thank the reviewer for the suggestion. We studied the recommended article and commented on the content accordingly in the discussion. The article is cited in the references.

The text and layout of Figure 1 may not be as intended.  Please review and improve for publication.  The text for the second box from the top does not fit the surrounding shape.

We thank the reviewer for his comment, we have corrected the appearance/text of Figure 1.

The conclusion includes: "COVID-19 likely represents a second hit of aHUS or TTP that manifests in certain part of genetically predisposed individuals.".  Please consider restating this as a hypothesis.  Likewise, please adjust similar statements in the article.  The article indicates candidate genes that can be examined in patients to evaluate this hypothesis; the text can be update to indicate evaluation options.

We thank the reviewer for the suggestion. To some extent, we have changed the expression and strength of the conclusion with the aim of non-determinism in a topic that is still new and poorly studied.

Reviewer 3 Report

Authors made all changes suggested by reviewers.

I have no more concerns.

Author Response

Dear reviewers,

thank you for the detailed review of our article and the given suggestions. We have done our best to take into account your comments and suggestions, which we believe have helped to improve the article.

Željka Večerić-Haler, the corresponding author

Reviewer 3

Authors made all changes suggested by reviewers.

I have no more concerns.

We would like to thank the reviewer for all previously given suggestions, which we believe had a significant impact on the quality of our paper